

# Dissolved and particulate organic carbon in Icelandic proglacial streams

Peter Chifflard[1], Christina Fasching[2a], Martin Reiss[1], Lukas Ditzel[1]

[1]Department of Geography, Philipps-University Marburg, Marburg, 35032, Germany
[2]Department of Limnology, University of Vienna, Vienna, 1090, Austria
[a]now at Department of Biology, Trent University, Peterborough, 1600, Canada

*Correspondence to*: Peter Chifflard (peter.chifflard@geo.uni-marburg.de)

**Abstract.** Here for the first time, we analyze the concentration of dissolved (DOC) and particulate organic carbon (POC), as well as its optical properties (absorbance and fluorescence) in proglacial streams of Iceland, location of Europe's largest
nonpolar ice cap. DOC and POC concentrations range from 0.11 mg L$^{-1}$ to 0.94 mg L$^{-1}$ and from 0.67 mg L$^{-1}$ to 173.33 mg L$^{-1}$, respectively. We estimate an annual release of 0.008 Tg C yr$^{-1}$ (DOC) and 1.72 Tg C yr$^{-1}$ (POC) from Icelandic glaciers. Compared to the global release of 1.97 Tg C yr$^{-1}$ POC, these first calculations underline the necessity to include the Icelandic glaciers in global organic carbon budgets. Based on optical properties, we found that although glacial derived organic matter (OM) was dominated by proteinaceous florescence, organic matter composition was variable among glaciers, often exhibiting
relatively higher aromatic content and increased humification closer to the glacier terminus, modulated by the presence of glacial lakes. While POC concentration decreased downstream, DOC concentration as well as the autochthonous fraction of OM increased suggesting the reworking of the OC by microbial communities, which has implications for downstream ecosystems as glaciers continue to melt.

**Keywords:** glacier, proglacial streams, dissolved and particulate organic carbon, Iceland






# 1 Introduction

Glaciers have only recently been recognized as unique ecosystems, with the potential to affect the carbon cycle. These systems accumulate organic carbon (OC) from the deposition of carbonaceous material derived from terrestrial and anthropogenic sources, and from in situ primary production (e.g., Anesio et al., 2009; Singer et al., 2012; Stibal et al., 2012; Hood et al., 

2015). The organic carbon stored in the glacial ice is released as dissolved and particulate organic carbon (DOC, POC), primarily through melt water at the glacier's surface, and subglacial flow discharges into proglacial streams and oceans (Bhatia et al., 2013). Microbial communities within these aquatic ecosystems can benefit from the partly high bioavailability of glacial DOC (Singer et al., 2012) and the export POC and associated nutrients such as phosphorus (Hodson et al., 2008). The POC component of glacial OC is considerable and can exceed twice the DOC concentration (e.g., Bathia et al., 2013). Recent global 

estimations by Hood et al. (2015) indicate an annual export of glacial DOC and POC of 1.04 TgC yr$^{-1}$ and 1.97 Tg C yr$^{-1}$, respectively. Current global projections forecast cumulative POC exports of 78 Tg by 2050, more than double the total predicted export of DOC (32.4 Tg) for the same period (Hood et al., 2015). However, currently POC data for glacier runoff is very limited and further studies are required to substantiate these current and future calculations (e.g., Bhatia et al., 2013; Hood et al., 2015).

Currently, there is a lack of regional analysis of the total amount of organic carbon released from glaciers. Estimations of the release of DOC and POC by Hood et al. (2015) are based only on 23 samples of the Antarctic Ice Sheet, 9 samples of the Greenland Ice Sheet and 55 of mountain glaciers. Furthermore, although the release of glacial organic carbon has been investigated in proglacial streams and in the glacial ice in Alaska (Spencer et al., 2014; Hood et al., 2015), the European Alps (Singer at al., 2012), Greenland (e.g., Bhatia et al., 2010; 2011; 2013; Lawson et al., 2014), Svalbard (e.g., Zhu et al., 2016), 

and Asia (e.g., Spencer et al., 2014), to our knowledge, there is no comparable information available for Iceland. Thus, Icelandic glaciers are not included in the global predictions of OC export (Hood et al., 2015), which is surprising as the largest nonpolar ice cap of Europe (Vatnajökull) is located in Iceland (Björnson et al., 2013). Existing studies including measurements of DOC and POC in Icelandic streams focus around: the impact of land degradation on carbon fluxes (Kardjilov et al., 2006), how damming impacts riverine fluxes to the ocean (both in northeastern Iceland; Eiriksdottir et al., 2017), and the transport of 

dissolved solids in glacier outburst floods – jökulhlaups (Galeczka et al., 2014; 2015). So far there is only one known study investigating DOC in Icelandic glacial ice — focusing on the diversity of snow algae (Lutz et al., 2015).

It is expected that climate-driven changes would have a greater impact on glacial runoff than other components of the hydrological cycle (Hood et al., 2015). Icelandic glaciers constitute 11 % of the land area and range in size from 3 km² (Gljúfurárjökull) to 8,100 km² (Vatnjajökull; Björnson et al., 2013). These temperate glaciers have been shown to be 

particularly sensitive to climatic fluctuations on an annual to decadal scale (Bradwell et al., 2013). While Icelandic glaciers show high accumulation rates during winter, they also show strong melting during summer (Mayer et al., 2017). In fact, it has been estimated that a total melting of all Icelandic glaciers would lead to a 1 cm rise in global sea level (Jóhannesson et al., 2013). Presently, glaciers account for approximately 30% of the all runoff (1,500 m³s$^{-1}$ of 5,000 m³s$^{-1}$) in Iceland (Björnsson





and Pálsson 2008; Jónsdottir 2008; Jónsdottir and Uvo 2009), highlighting the important hydrological role of glaciers in Iceland. Since the mid-1990's Icelandic glaciers have shown an average annual mass loss of $9.5 \pm 1.5$ Gt a$^{-1}$, resulting in a total loss of 84 km³ from the icecap volume (e.g., Jaenicke et al., 2006; Jóhannesson et al., 2013; Mayer et al., 2017).

Due to the altitude differences and the geographical position of the glaciers within Iceland, at the border between Arctic and temperate seas, and the cold air masses of the Arctic and warm air masses of lower latitudes, Icelandic glaciers show varying mass balances depending on their position (Björnsson et al., 2013). For example, over the period 1995 to 2010 Vatnajökull lost 3.7% of its total ice mass, while Hofsjökull and Langjökull both lost 11%. Furthermore, the ice cap Snæfellsjökull has lowered by 14 m on average between 1999 and 2008, corresponding to an average mass loss of 1.25 m$_{w.e.}$ per year (Jóhannesson et al., 2011). Model prediction for the future development of the mass balances to the year 2040 estimate a retreat of 25% of the current volume for Hofsjökull and Vatnajökull (Björnsson and Pálsson 2008; Aðalgeirsdóttir et al., 2006). Particularly in the latter case, the southern outlet glaciers are vulnerable to warming and climate change (Aðalgeirsdóttir et al., 2006; Bradwell et al., 2013). At Langjökull, estimated volume reduction will be more intensive, - 35% of the total volume, during the same time period due its lower elevation in comparison to Hofsjökull and Vatnajökull (Pope et al., 2016). By the year 2190 Langjökull will have disappeared totally, while only 5% of the total volume of the higher elevation Hofsjökull will remain (Guðmundsson et al., 2009). Considering these spatial variations within the fast melting process, meltwater runoff is expected to initially increase, peaking in the years around 2030, followed by a constant decrease of annual runoff in the following 100 years (Bliss et al., 2014; Björnsson and Pálsson 2008). These temporal patterns differ greatly from other glaciers, such as those in the Canadian or Russian Arctic which are expected to show peak runoff around 2080. This is a result of Iceland's geographical position, situated in the climatically sensitive boundary area between polar and mid-latitude atmospheric circulation cells and higher mean summer air temperatures projected for the period 2003-2022 and 2081-2100 (Iceland: +5.1°C/6.5°C; Russian arctic: +0.8°C/+2.8°C; Canadian arctic: +0.4°C/+1.9°C) (Björnsson et al., 2013; Bliss et al., 2014). Nevertheless, apart from studies investigating the impact of runoff changes on hydropower, no information about the release of DOC and POC of Icelandic glaciers, and the corresponding concentrations in proglacial streams are available. Galeczka et al., (2014; 2015) measured DOC and POC concentration during two jökulhlaups, but there exist no comparable measurements during normal flow conditions.

While compared to non-glaciated catchments, the exported amount of OC of glacier ecosystems is typically low (Mulholland 1997; Barker et al., 2006; Bathia et al., 2010), exports support heterotrophic metabolism in proglacial streams, as the bioavailability of glacier derived OC is noticeably higher (BDOC, $59\pm20\%$; Singer et al., 2012) than those from headwaters in forested catchments ($25\pm5\%$; Volk et al., 1997) or in structured natural catchments including forests, agricultural soils or wetlands (mean 25%; Risse-Buhl et al., 2013). Moreover, compared to temperate catchments, there are relatively less terrigenous OC inputs from the riparian zone or organic soils in glaciated catchments (Castella et al., 2001; Gíslason et al., 1998; 2000; 2001). Consequently, the composition and bioavailability of glacial derived carbon plays an important role for the aquatic food web and for carbon cycling in the proglacial streams (Gíslason et al., 2001, Fellman et al., 2015). In contrast to Greenland, several proglacial streams in Iceland have a long flowpath, from the glacial terminus to the Atlantic Ocean, which



may result in the input of additional terrestrial DOC inputs from the increasingly unglaciated downstream catchment area. This may impact the composition and bioavailability of OC along these streams (Milner et al., 2017; Shirokova et al., 2017). However, the composition and concentration of organic carbon in proglacial streams which is released from Icelandic glaciers remain unknown.

Here we investigate the concentration and composition of glacial derived OC of Icelandic glaciers for the first time. We assess: i) the concentration of dissolved and particulate organic carbon (DOC and POC), ii) DOM composition (as absorbance and fluorescence) in several Icelandic proglacial streams and iii) the longitudinal changes of DOC and POC concentrations, as well as composition along a proglacial stream from the glacier terminus to the Atlantic Ocean. Our study aims to provide insight into the concentration and composition of OC in ~13 different Icelandic proglacial streams, estimating Icelandic glacial

OC fluxes for the first time. Additionally, this study contributes to an enhanced worldwide prediction of OC export from glaciers as there is no comparable data with such a high spatial resolution available for Iceland.

## 2 Materials and Methods

### 2.1 Sampling points

We sampled stream water from 25 sites, draining a total of 5 Icelandic glaciers during the melting season (23-31 July 2016).

The melting season was chosen as during this period the ablation zones of the glaciers are free of snow and the proglacial streams cover OC of different meltwater sources (supraglacial, englacial and subglacial) according to the findings of Bhatia et al. (2011; 2013) and Das et al. (2008). During the summer melt season, the main water source areas of glacial organic carbon are linked by a hydrological network within the glacial ecosystem, with the characteristics of the melt water in the proglacial streams reflecting the contribution of these sources (Bhatia et al., 2011; 2013; Das et al., 2008). Sampling points covered melt

water from glaciers of the Icelandic ice caps Vatnajökull, Langjökull, Hofsjökull, Myrdalsjökull, and Tungnafellsjökull (11 samples), and samples of non-glaciated streams for comparison (2 samples) (Fig. 1, Table 1). In order to assess the riverine transformation of organic carbon, a further 11 water samples were taken along the river Hvitá, starting at the glacier Langjökull and ending at its terminus at the Atlantic Ocean in the southwest of Iceland.

### 2.2 Sample preparation and field parameters

Electrical conductivity, water temperature and pH of the unfiltered meltwater was measured using a regularly calibrated portable water quality meter (Hanna Combo HI98129). For further analyses (POC, DOC and optical analysis) water samples (150 ml) were filtered using pre-combusted (4h at 450°C) glassfibre filters (Whatman GF/F, pore size 0.7 µm) according to Singer et al. (2012) and stored in a cooling box in glass vials (acid washed and combusted at 450°C for 4h) until shipment and laboratory analysis.



## 2.3 Analysis of DOC and POC concentration

DOC concentrations were measured using a TOC analyzer (TOC-L, Shimadzu, Japan) using high-temperature combustion of organic matter (OM) followed by thermal detection of $CO_2$. POC was measured by determining mass lost upon combustion of the samples. The glass fiber filters were dried after sampling at 65°C to a constant weight to determine the total suspended

solids (TSS). The samples were then combusted at 550° C and reweighed to calculate the amount of particulate organic matter according to Maciejewska and Pempkowiak (2014).

## 2.4 Absorbance measurements, Excitation emission matrices (EEMs) and parallel factor analysis (PARAFAC)

The excitation emission matrices (EEMs) were generated by measuring fluorescence intensities at excitation wavelengths ranging from 200 to 450 nm (5 nm increments) and emission wavelengths from 250 to 700 nm (2 nm increments) with a scan

speed of 12,000 nm $min^{-1}$ according to Singer et al. (2012). We used a 1 cm quartz cuvette and a fluorescence spectrometer (Shimadzu RF-6000) for analysis. Absorbance was measured using a UV–VIS spectrophotometer (Genesys 10S, ThermoFisher) and 1-cm quartz cuvettes. Absorbance and fluorescence measurements were conducted at the Department of Geography, Philipps-University of Marburg, and used to calculate a suite of indices, such as the slope ratio – as an indicator of molecular weight ($S_R$, Helms et al., 2008), as well as the specific UV absorbance at 280 nm (SUVA280; Chin et al., 1994)

– indicating aromaticity. In addition, the humification index (HIX; Zsolnay et al., 1999), the freshness index (ß/α; Parlanti et al., 2000; Wilson and Xenopoulos 2009) and fluorescence index (FI; McKnight et al., 2001) were used to determine the degree of humification, autochthonous input, and DOM source, respectively. Based on excitation emission matrices (EEM) generated by measuring the fluorescence of DOM (Shimadzu RF-6000), parallel factor analysis (PARAFAC, Stedmon and Bro 2008) was used to model individual fluorescent components and to detect the origin of glacial DOM (Fig. 1). Modeling was

performed in Matlab (7.11.0) using the DOMFluor Toolbox (1.7; containing the N-Way toolbox, 3.1; Andersson and Bro 2000).

## 3 Results and Discussion

### 3.1 DOC and POC concentration in the proglacial streams

Measured DOC-concentrations were relatively low, ranging from 0.11 mg $L^{-1}$ to 0.94 mg $L^{-1}$  (Table 2), but closely bracketed

by values measured in the melt water for comparable regions such as the Greenland Ice Sheet (average: 0.51 mg $L^{-1}$), Antarctic Ice Sheet (0.43 mg $L^{-1}$) and mountain glaciers (0.37 mg $L^{-1}$) (Hood et al., 2015). POC concentrations ranged from 0.67 mg $L^{-1}$ to 173.33 mg $L^{-1}$ and exceeded their respective DOC concentrations at every sampling location. This observed relationship between DOC and POC is comparable to the Greenland Ice Sheet, but contrasts small glaciers in the European Alps, where the concentrations of DOC and POC are more or less equal (Hood et al., 2015). Nevertheless, studies citing comparable POC

values are limited (Hood et al., 2015). The highest POC concentration was measured at the terminus of Sólheimajökull (JK01),

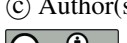



despite the presence of a glacial lake which is located between the outlet of the glacier Sólheimajökull and the sampling point, possibly causing sedimentation of organic and inorganic material. The size of the lake may play an important role in POC sedimentation, as samples taken downstream of the larger lakes Sandvatn (HV07) and Hvítárvatn (HV10), showed lower POC concentrations (0.67 mg L$^{-1}$ and 4.0 mg L$^{-1}$ respectively). Glacial melt water from the Skaftafellsjökull (SK01) and the

Svínafellsjökull (SV01) displayed similar DOC and POC concentrations of 0.15 mg L$^{-1}$ and 40 mg L$^{-1}$ and 0.14 mg L$^{-1}$ and 46 mg L$^{-1}$, respectively. In contrast, the two main rivers draining the Súlujökull and Skeidarárjökull (sampling points: SJ01, SJ02) showed different DOC and POC concentrations, which suggests the influence of local factors such as: slope, glacier size, melt water temperature, or differences within the glacial catchment. Additionally, the POC concentration at point SJ01 was higher than at the other three sampling points. Thus, proglacial lakes may have a significant effect, on POC concentrations at the

points SJ02, SV01 and SK0 due to the presence of the larger proglacial lake between the glacier terminus and sampling point in comparison to point SJ01. These findings highlight the role of proglacial lakes for biogeochemical processes, with potential consequences for the aquatic systems and the planktonic organisms inhabiting them (Sommaruga 2015).

DOC and POC concentrations of proglacial streams differed between the individual glaciers. Higher DOC concentrations were

measured in the proglacial streams originating from Hofsjökull (AJ01, HN01) and Langjökull (BL02), with values ranging from 0.16 mg L$^{-1}$ to 0.26 mg L$^{-1}$. DOC concentrations of the proglacial streams, related to the main glacier (Mýrdalsjökull: JK01, MV01; Langjökull: e.g. BL02, HV10, HV09; Tungnafellsjökull: TJ01, TJ02; Vatnajökull: SK01, SJ01, SJ02, SV01), were found to be lower (0.11 mg L$^{-1}$ to 0.16 mg L$^{-1}$). POC concentrations of the proglacial streams draining Mýrdalsjökull and Vatnajökull peaked at 173 mg L$^{-1}$. Such elevated concentrations may have implications for biogeochemical processes in the

ocean, due to the relatively short distance from the glacier terminus to the North Atlantic Ocean compared to other glaciers (Bhatia et al., 2013). Therefore, the majority of OC may be transported to the coastal zone without intensive processing within the proglacial stream (e.g. burial, microbial reworking, outgassing etc.) or retained in reservoirs like in the northeast of Iceland (Eiriksdottir et al., 2017). This increased input of organic carbon of Mýrdalsjökull and Vatnajökull has to be taken into account since terrestrial organic carbon, especially from glacial runoff, represents an important source of carbon to near-shore coastal

areas (Fellmann et al., 2010; Bhatia et al., 2013).

**3.2 DOC and POC concentrations along the river Hvitá**

Both POC and DOC quantity differed longitudinally along the river Hvitá from the proglacial lakes Hagavatn and Hvítárvatn toward the river mouth at the Atlantic Ocean nearby Eyrarbakki (Figure 2). DOC concentrations at the outlet of both lakes were similar, whereas the POC concentration differed, with the lower concentrations at HV10 likely caused by sedimentation

in the larger and longer lake Hvítárvatn. Along the flowpath of the rivers Hvitá (HV11, HV05) and Tungufljiot (HV07, HV06, HV04), DOC concentration remained relatively constant, indicating limited input of further OC from the surrounding catchment, likely due to sparse or complete lack of vegetation (e.g., Gíslason et al., 2001). This underlines the importance of glacier derived organic carbon inputs to the proglacial streams in the upper reaches. Downstream, the vegetation cover of the





catchment increased along with anthropogenic impact. This possibly lead to elevated DOC concentrations as we measured the highest DOC concentration near an urbanized location – Selfoss, HV02 (0.94 mg L$^{-1}$), underlining the impact of urbanized areas on water quality and DOC (e.g. Hatt et al., 2004). At the sampling point HV01, located at the end of the lagoon Ölfusá, in the estuary of the river Hvitá, the DOC concentration decreased, likely due to the strengthened influence of the seawater, as

indicated by higher electrical conductivity (Table 2). Additionally, low flow velocities in the lagoon may induce sedimentation and cause lower POC concentrations. Fellman et al. (2010) highlighted the role of estuaries as a critical link for the transport of DOC and POC between terrestrial and marine ecosystems. Similarly, the role of lagoons which are typical for the southwest coast of Iceland have to be taken into account in term of the export of OC into the ocean.

### 3.3 Composition and spatial variability of DOM in Icelandic proglacial rivers

Based on the generated excitation emission matrices (EEM) and using PARAFAC, two individual fluorescent components were modeled (Fig. 3). The first component resembled protein-like fluorescence of tyrosine (Coble 1996; Stedmon and Markager 2008) and tryptophan, which are prominent contributors to total DOM fluorescence in various glaciers (Barker et al., 2006; Dubnick 2010). These are likely derived from algae and bacteria, as often observed in glacial environments (Hodson et al., 2008). The second component resembled humic-like fluorescence of terrestrial/allochthonous origin, comparable to

peak A found by Coble (1996). We acknowledge that our comparatively small sample size adds some degree of uncertainty to the PARAFAC results, and likely does not capture the full diversity of fluorophores in these environments. However, the high percentage of C2 (47-92%) points to proteinaceous DOM as a main contributor to glacial DOM, which is in line with previous studies (Hodson et al., 2008; Hood et al, 2009).

To assess the spatial variability of DOM within the sampled proglacial streams we computed a principal component analysis (PCA), based on the optical indices and fluorescent components of the proglacial streams (Figure 4). Although glacial DOM generally exhibited high proteinaceous fractions (C2), the PCA revealed DOM properties to vary among glaciers, with closer sampling points being more closely related in terms of DOM optical composition. Glacial derived DOM from Langjökull was composed of relatively fresher DOM (higher β/α), with a higher contribution of the protein-like component C2, while glacial

derived DOM from Myrdalsjökull contained relatively more autochthonous material (higher contributions of C1 and higher HIX). However, the sampling points closest to the glacier termini showed distinct optical properties, differing from the other sampling points in the same region (Fig. 5). This may be partially due to increased residence times in the proglacial lakes at the glacier terminus. In fact, the sampling points TJ01, SV01 and JK01 are all close (1-3 km) to the terminus of the glaciers Tungnafellsjökull, Svinafellsjökull and Sólheimajökull, but the DOM composition is very variable (Figure 4). Meltwater closer

to the terminus of the glaciers Tungnafellsjökull and Sólheimajökull exhibited a relatively more allochthonous character, possibly pointing to ancient vegetation as the OC source. Although Hood et al. (2009) found that elevated contributions of glaciers in a catchment increase the fraction of proteinaceous DOM in proglacial streams, Singer et al. (2012) found phenolic compounds derived from vascular plants or soil dominate the highly diverse glacial derived DOM. This indicates that glacial

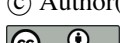


melt exhibits a distinct DOM composition which is subject to microbial metabolism, possibly explaining downstream compositional changes (Guillemette and del Giorgio, 2012).

POC concentrations exhibited a distinct spatial pattern, with highest values in proglacial streams originating from the glaciers Vatnajökull, Myrdalsjökull and Hofsjökull, and lower values in Langjökull. POC represented the quantitatively more important fraction of OC. We found POC to decrease, while DOC increased with the distance from the glacier termini (Figure 6). In fact, POC concentrations were highest close to the glacier termini, especially compared to streams draining unglaciated areas. The decrease in POC may be due to direct use and reworking by benthic organisms (Findlay, 1995). At the same time, DOM composition changed from relatively more allochthonous to autochthonous DOM. This may be the result of recent autochthonous in-stream production, likely from algae and or microbial reworking of the material (Kaplan and Bott, 1989).

Optical DOM characteristics along the river Hvitá revealed that sampling points at the lower part of the river had similar characteristics (e.g. HV01, HV02, HV03, HV05) (Figure 4), with particularly the proteinaceous component (C1) showing high intensities along the flowpath (Figure 6). A changed DOM composition became recognizable at the sampling points HV07, HV09 and slightly at HV10 which are located in the headwater of the river Hvitá, – at shorter distances to the glacier terminus (Table 1). This indicates that glacial derived organic carbon is processed (microbial) downstream, but the compositional changes are recognizable at first after a distance of approximately 60 km (around sampling point HV05). Consequently, glacial derived OC in proglacial streams with a short distance (approx. 2 to 9 km) from the glacier terminus to the Atlantic Ocean is subject to comparatively less alteration, and directly serves as a source of carbon and nutrients for marine aquatic heterotrophs. However, the influence of the influx of glacial derived and riverine organic matter on the trophodynamics of coastal marine food webs is still not well understood (Fellman et al., 2010; Arimitsu et al., 2017).

## 4 Conclusions

For the first time we investigate the spatial variability of DOC and POC in proglacial streams in Iceland, expanding the current understanding and ability to predict global glacial OC export and composition. Although the Icelandic glaciers Vatnajökull, Langjökull, Hofsjökull, Myrdalsjökull and Tungnafellsjökull are DOC-poor ecosystems, our findings highlight the role of glacier derived OC for downstream carbon fluxes. Along the first 60 km along the river Hvitá DOC concentration was substantially unchanged, suggesting input of further OC from the surrounding catchment is limited due to the sparseness or complete absence of vegetation. DOC-concentrations ranged between 0.11 and 0.94 mg L$^{-1}$ and are comparable with values in other regions (e.g., Alaska, Greenland, European Alps), while POC concentrations (up to 173.33 mg L$^{1}$) exceeded those of the DOC concentration in every investigated proglacial stream.

Assuming our results are representative, we estimate an annual release of 0.008 Tg C yr$^{-1}$ (DOC) and 1.72 TgC yr$^{-1}$ (POC) from Icelandic glaciers, based on a mean runoff of 1,500 m³ s$^{-1}$ (Björnsson and Pálsson 2008) from Icelandic glaciers, and using the mean concentration of DOC and POC of sampling points with distances less than 20 km to the glacier terminus



(n=13). We excluded sampling point JK01 (POC: 173.33 mg/l) from our calculation due to potentially significant anthropogenic influence at this site. Our generally higher POC values may be the result of the dehydration of hydrated clay minerals, present in volcanic ash and soils such as smectites, which contain large quantities of water of hydration, which is expelled at ~300° (Lagaly 1993). This loss of the water of hydration upon combustion of the filters (up to 550°C) may account

for the potential overestimation of POC concentration at some sites. Thus, for further studies determining POC in Icelandic proglacial streams, the application of a C/N-analyzer is recommended. Nevertheless, compared to the global release of 1.97 Tg C yr$^{-1}$ POC estimated by Hood et al. (2015), these first calculations underline the absolute necessity to include the Icelandic glaciers in global organic carbon budgets. Moreover, if the annual release of DOC is weighted by the glaciated area of Iceland (11,060 km²; Grinstedt 2013), the calculated value is 0.0007 Gg C yr$^{-1}$ km$^{-2}$, clearly exceeding the area-weighted estimations

of the Greenland Ice Sheet and the European Alps (0.0002 Gg C yr$^{-1}$ km$^{-2}$ each) (Hood et al. 2015; Grinstedt 2013).

Although DOM composition was highly variable, it was generally dominated by proteinaceous fluorescence, and often changed distinctly downstream of the glacier terminus, possibly due to the presence of proglacial lakes. Sampling points closer to the termini exhibited a distinct composition, often also dominated by allochthonous DOM. While POC concentration

increased downstream, DOC concentration as well as its autochthonous fraction increased, pointing to reworking of the OC by microbial communities, as well as benthic organisms, and autochthonous production of DOM by algae. Additionally, the observed longitudinal changes of DOC concentration along the proglacial stream Hvitá highlights the importance of glacial derived OC for downstream carbon balance. Urban settlements (near HV02) led to a spike in DOC concentration (up to 0.94 mg L$^{-1}$), while along the length of the stream (80 km) between HV10 (most upstream sampling point) and HV03 (upstream of

the settlement) DOC concentration only increased from 0.15 mg L$^{-1}$ to 0.34 mg L$^{-1}$.

According to recent predictions, an increase of glacial runoff can be expected, which will considerably impact the input of glacial derived OC to proglacial streams (Hood and Berner, 2009, Bliss et al., 2014). Our findings reveal a first and important insight into the spatial variability of DOC and POC concentration, as well as the DOM composition in Icelandic proglacial

streams. In contrast to the Greenland ice sheet, the predicted melt rates for Icelandic glaciers are distinctly higher, thus the release of OC stored in Icelandic glaciers is important for the surrounding environment with implications for downstream ecosystems and associated species. In order to further constrain current predictions of the OC export due to the rapid melting of Icelandic glaciers, further samples of both proglacial streams and glacial ice are required.






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





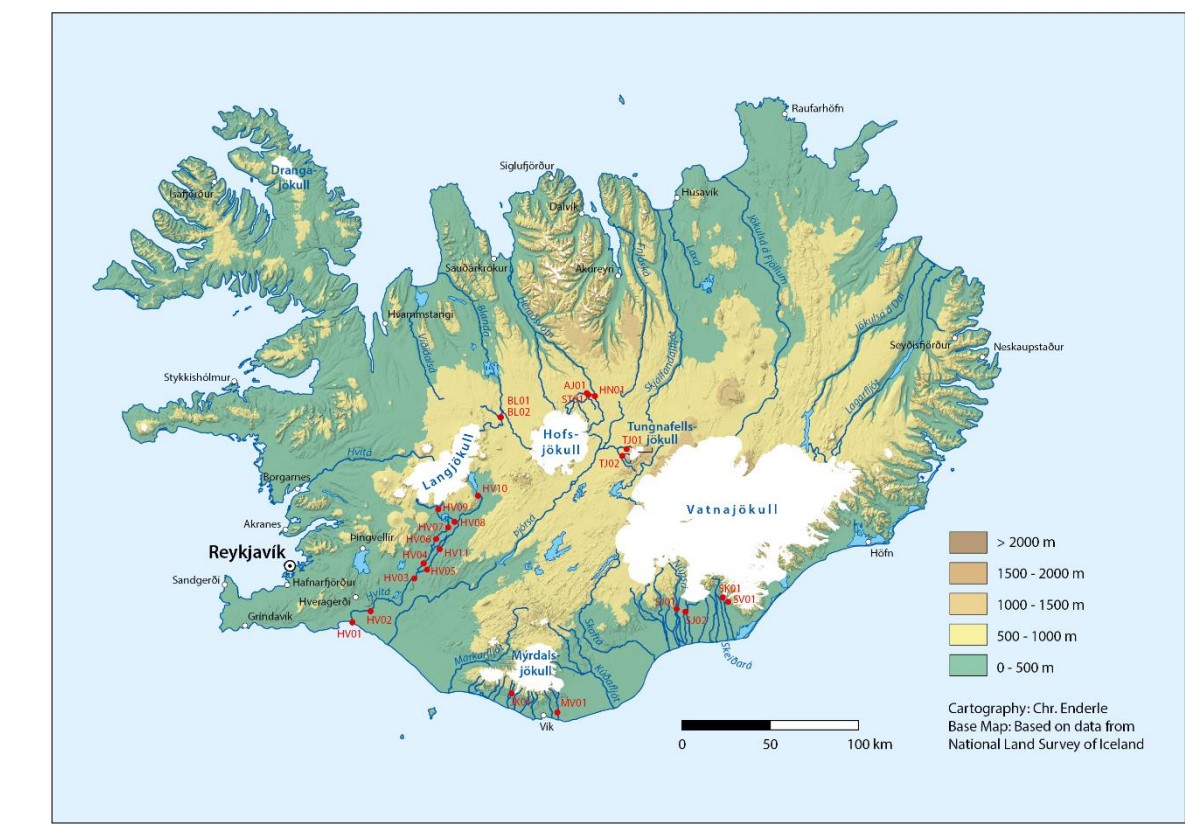

Figure 1: Topography of Iceland with glacier distribution and the location of sampling points.





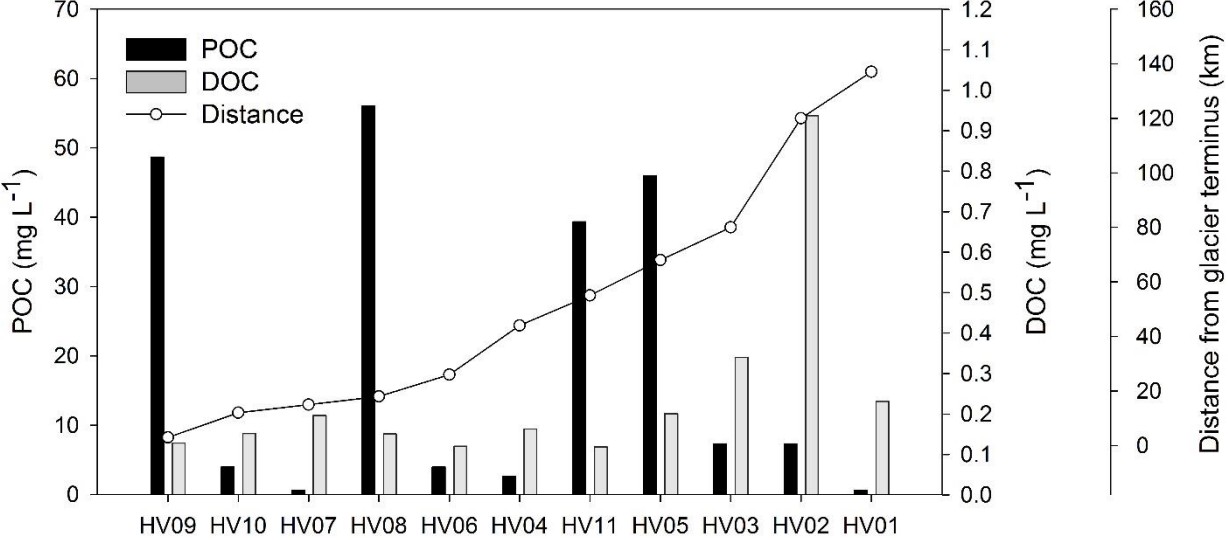

**Figure 2: DOC and POC concentrations of water samples along the river Hvitá (HV01, HV02, HV03, HV08, HV10, HV11) and the tributaries Tungufljiot (HV04, HV06, HV07, HV09) and Sandá (HV08). Shown as points is the distance of each sampling location from glacier termini in kilometers.**





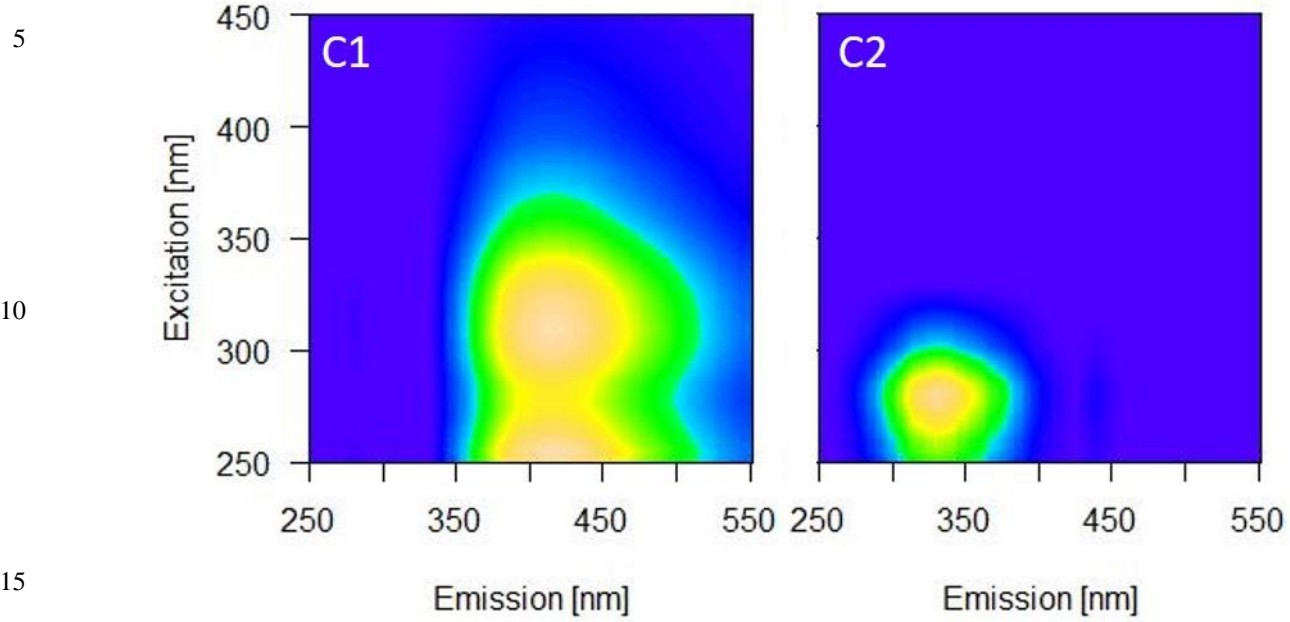

**Figure 3: Fluorescent components of glacial DOM modeled by PARAFAC based on excitation emission matrices. Samples were taken during the field campaign in summer 2016.**

25

30



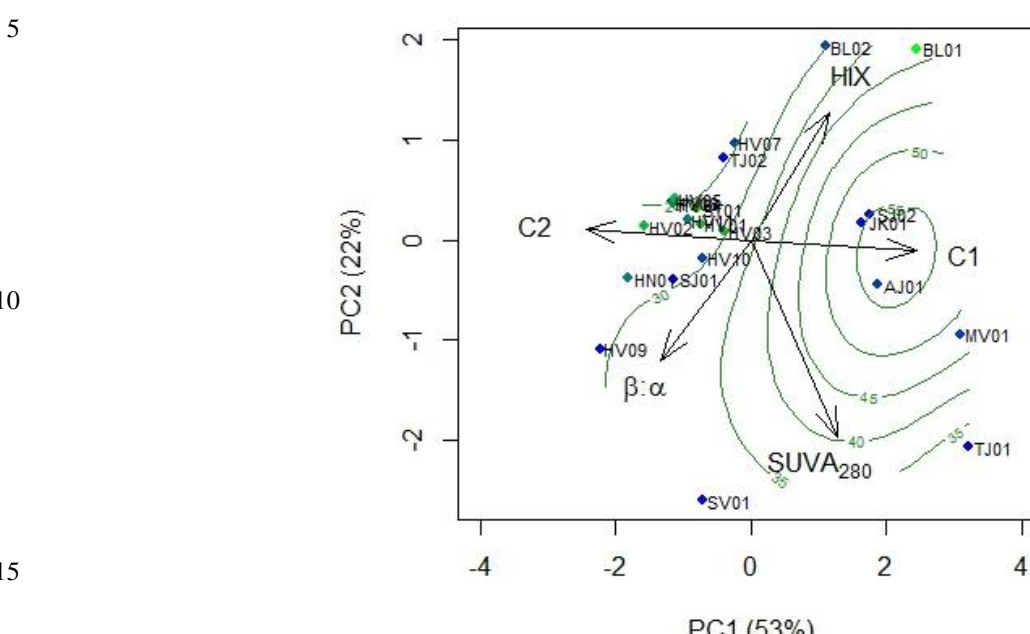

**Figure 4: Principal component analysis (PCA) based on the optical properties of the proglacial streams distinguishes terrigenous from autochthonous DOM (PC1) (n=21). Terrigenous DOM was characterized by a high humification index (HIX), higher SUVA$_{280}$ and higher contributions of the humic-like component (C1). High values of the freshness index ($\beta/\alpha$) and high contributions of the protein-like component (C2) describe DOM with a more autochthonous character. Color of the symbols represents distance from glacier terminus, with green symbols values representing sampling points closer to the terminus, while blue represent points further downstream and samples from unglaciated catchments (BL01 and ST01). Arrows are based on PCA structural coefficients and contours represent POC concentration in mg L$^{-1}$.**





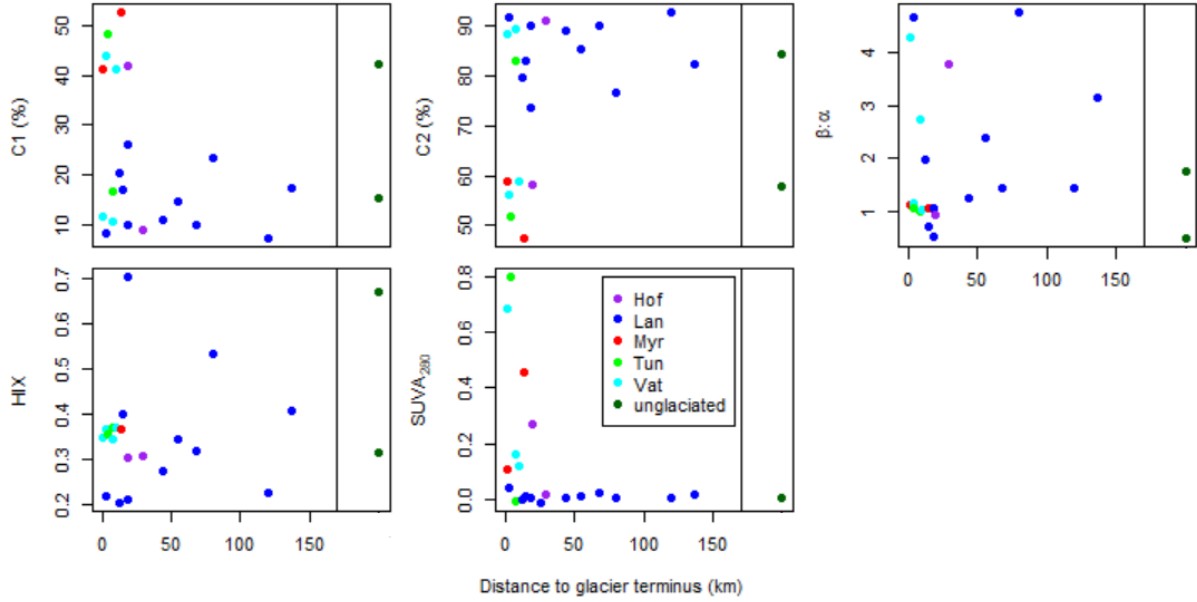

**Figure 5: Change of humification index (HIX), SUVA₂₈₀, freshness index (β/α) and components C1 and C2 with distance from each sampling point to the glacier terminus. Color indicates the respective glacier (Vat=Vatnajökull, Lan = Langjökull, Hof = Hofsjökull,** 5 **Myr = Myrdalsjökull, and Tun = Tungnafellsjökull) and unglaciated area (un).**



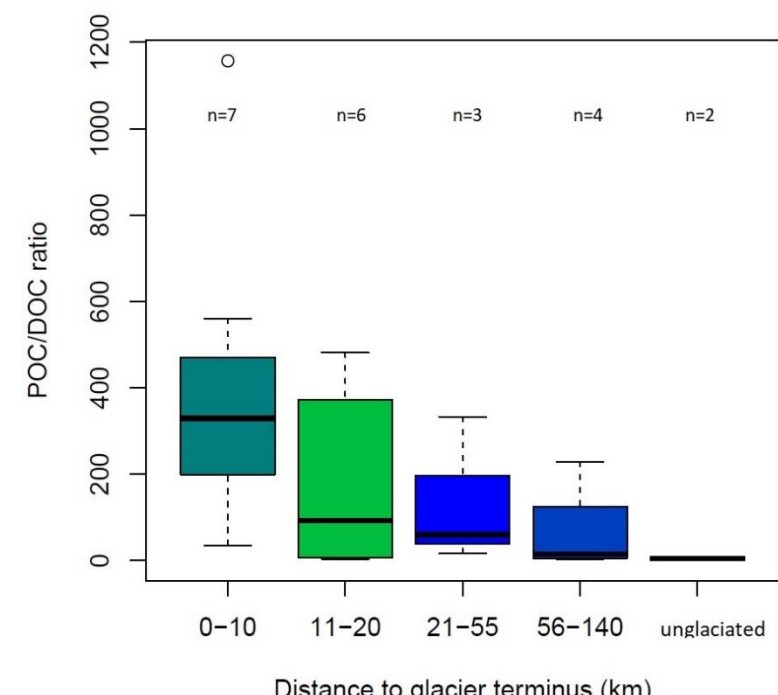

**Figure 6: Boxplot of the POC/DOC ratio along distance classes from glacier termini. Color represents DOM composition (PC1), with green representing relatively more allochthonous inputs and blue more autochthonous DOM (n=21).**





**Table 1: Description of the sampling points.**

| Site | Main Glacier | Sub Glacier | Sampling time (GMT) | Coordinates | Distance to glacier [km] |
|------|--------------|-------------|---------------------|-------------|--------------------------|
| HV01 | Langjökull | Hagafellsjökull/Nordurjökull | 24.07.2016 (18:00) | N 63° 52.710 W 021° 12.720 | 137 |
| HV02 | Langjökull | Hagafellsjökull/Nordurjökull | 24.07.2016 (18:30) | N 63° 56.295 W 021° 00.350 | 120 |
| HV03 | Langjökull | Hagafellsjökull/Nordurjökull | 24.07.2016 (19:30) | N 64° 06.646 W 020° 30.660 | 80 |
| HV04 | Langjökull | Hagafellsjökull | 25.07.2016 (11:15) | N 64° 11.290 W 020° 24.320 | 44 |
| HV05 | Langjökull | Nordurjökull/Hagafellsjökull | 25.07.2016 (11:00) | N 64° 09.402 W 020° 21.890 | 68 |
| HV06 | Langjökull | Hagafellsjökull | 25.07.2016 (11:50) | N 64° 18.825 W 020° 15.945 | 26 |
| HV07 | Langjökull | Hagafellsjökull | 25.07.2016 (13:15) | N 64° 22.380 W 020° 07.645 | 15 |
| HV08 | Langjökull | Hagafellsjökull | 25.07.2016 (14:15) | N 64° 24.125 W 020° 03.210 | 18 |
| HV09 | Langjökull | Hagafellsjökull | 25.07.2016 (15:20) | N 64° 27.930 W 020° 14.800 | 3 |
| HV10 | Langjökull | Nordurjökull | 25.07.2016 (17:00) | N 64° 32.200 W 019° 46.860 | 12 |
| HV11 | Langjökull | Nordurjökull/Hagafellsjökull | 25.07.2016 (12:30) | N 64° 15.725 W 020° 13.350 | 55 |
| BL01 | Unglaciated drainage area | | 26.07.2016 (09:45) | N 64° 56.216 W 019° 31.280 | |
| BL02 | Langjökull | Baldkökull | 26.07.2016 (10:00) | N 64° 56.200 W 019° 31.250 | 18 |
| AJ01 | Hofsjökull | Illvidrajökull | 26.07.2016 (16:30) | N 65° 03.530 W 018° 29.300 | 19 |
| ST01 | Unglaciated drainage area | | 26.07.2016 (17:00) | N 65° 03.000 W 018° 28.200 | |
| HN01 | Hofsjökull | Miklafellsjökull/Klakksjökull | 26.07.2016 (17:30) | N 65° 02.630 W 018° 23.460 | 29 |
| TJ01 | Tungnafellsjökull | Hagajökull | 27.07.2016 (10:20) | N 64° 46.350 W 018° 01.200 | 4 |
| TJ02 | Tungnafellsjökull | Tungnafellsjökull | 27.07.2016 (11:00) | N 64° 44.300 W 018° 04.330 | 8 |
| JK01 | Mýrdalsjökull | Sólheimajökull | 28.07.2016 (15:00) | N 63° 32.050 W 019° 22.290 | 1 |
| MV01 | Mýrdalsjökull | Kötlujökull | 28.07.2016 (16:10) | N 63° 26.250 W 018° 51.100 | 14 |
| SJ01 | Vatnajökull | Súlujökull | 28.07.2016 (18:00) | N 63° 57.350 W 017° 28.160 | 8 |
| SJ02 | Vatnajökull | Skeidarárjökull | 28.07.2016 (18:20) | N 63° 56.390 W 017° 22.150 | 10 |
| SV01 | Vatnajökull | Svinafellsjökull | 29.07.2016 (09:00) | N 63° 59.025 W 016° 52.390 | 1 |
| SK01 | Vatnajökull | Skaftafelljökull | 29.07.2016 (10:20) | N 64° 00.390 W 016° 56.035 | 3 |



**Table 2: DOC and POC concentrations and hydrochemical characteristics of the sampled Icelandic proglacial rivers.**

| Site | pH [-] | Electr. Conductivity [$\mu$S cm$^{-1}$] | Water temperature [°C] | DOC [mg L$^{-1}$] | POC [mg L$^{-1}$] |
|------|--------|------------------------------------------|-------------------------|---------------------|---------------------|
| HV01 | 7.9 | 300.0 | 14.4 | 0.23 | 0.67 |
| HV02 | 8.6 | 55.0 | 14.5 | 0.94 | 7.33 |
| HV03 | 7.8 | 102.0 | 15.3 | 0.34 | 7.33 |
| HV04 | 8.3 | 40.0 | 8.9 | 0.16 | 2.67 |
| HV05 | 8.3 | 51.0 | 12.9 | 0.20 | 46.00 |
| HV06 | 9.2 | 34.0 | 7.2 | 0.12 | 4.00 |
| HV07 | 10.0 | 21.0 | 8.7 | 0.20 | 0.67 |
| HV08 | 9.1 | 12.0 | 14.5 | 0.15 | 56.00 |
| HV09 | 8.5 | 2.0 | 10.3 | 0.13 | 48.67 |
| HV10 | 8.9 | 42.0 | 13.4 | 0.15 | 4.00 |
| HV11 | 8.6 | 48.0 | 11.9 | 0.12 | 39.33 |
| BL01 | 9.7 | 53.0 | 9.4 | 0.56 | 0.00 |
| BL02 | 10.2 | 47.0 | 9.9 | 0.22 | 1.33 |
| AJ01 | 9.6 | 32.0 | 9.6 | 0.26 | 41.33 |
| ST01 | 9.5 | 38.0 | 11.5 | 0.29 | 2.67 |
| HN01 | 9.7 | 22.0 | 10.9 | 0.16 | 9.33 |
| TJ01 | 10.6 | 20.0 | 5.7 | 0.11 | 4.00 |
| TJ02 | 11.1 | 27.0 | 9.8 | 0.14 | 20.00 |
| JK01 | 10.9 | 48.0 | 5.6 | 0.15 | 173.33 |
| MV01 | 10.4 | 160.0 | 7.2 | 0.15 | 70.00 |
| SJ01 | 10.3 | 18.0 | 9.3 | 0.15 | 84.67 |
| SJ02 | 10.7 | 35.0 | 4.6 | 0.22 | 55.33 |
| SV01 | 12.3 | 32.0 | 2.8 | 0.14 | 46.00 |
| SK01 | 13.1 | 14.0 | 3.7 | 0.15 | 40.00 |