# Peer review of "Dissolved and particulate organic carbon in Icelandic proglacial streams"

_The Cryosphere, 2018_

## Referee Comment (RC1) · Anonymous Referee #1 · 10 Apr 2018

General comments: This study measures the concentration and flux of dissolved and particulate organic carbon (OC) over a week long period from proglacial streams in Iceland. This study further uses spectroscopic characterization of DOM to assess how its composition varies across the study streams. The authors found that Icelandic glaciers are releasing large quantities of DOC and especially POC compared to previous global estimates of organic carbon release from glaciers. Overall, this is an interesting study that could potentially fill a key information gap in our understanding of how organic carbon release from glaciers varies across the globe. However, as written, there are far too many methodological uncertainties/limitations for consideration of publication. For instance, a great deal more information is needed for sample processing and analysis, OC flux estimates, and assessment of analytical error. It appears that one sample was

collected one time from each stream, with no assessment of analytical error. Moreover, my largest concern lies with the POC analyses. As the authors state in the conclusions (should be stated in the methods), POC concentrations are not measured directly but rather by loss on ignition. First off, this is not a very commonly used technique for POC analysis, especially in stream biogeochemistry. Second, the authors clearly acknowledge that hydrated clay minerals are likely contributing to the high POC concentrations. Third, it is clear that the POC concentrations are highly variable and greatly overestimated. How can concentrations increase from <1 to 56 mg/L over a distance of 3km in river length? Without any replicates or lab studies comparing direct POC vs. LOI analyses, I don't know how these data can be compared to other glacier OC studies with direct POC analysis.

What follows are more specific comments for the authors to consider to help improve clarity and interpretation of findings.

More specific comments: Page 2, lines 20-22: This is confusing. Icelandic glaciers are included in the global estimates of DOC and POC release from glaciers. However, as the authors point out, concentration estimates of OC in Icelandic glaciers as well as many other regions worldwide are not. Please rephrase to improve clarity.

Page 4, line 10: I think the authors have already made it clear that there are no other studies, to their knowledge, of OC release from Iceland glaciers.

Page 4, line 14: This is a very short sampling window. How long does the melt season last? When is the peak of the melt season? How many times was each stream sampled?

Page 4, lines 26-29: Given the extremely low DOC concentration reported here, more information is needed on sample handling and processing. For instance, were samples field filtered or acidified in the field? Were replicate samples collected? How long were sampled stored in the field before transport? How long until analysis for DOC occurred?

Page 5, lines 1-6: More information is needed about the DOC and POC analysis, such were replicated analyses performed? What is the lower detection for DOC? Are there any error estimates on the OC concentrations? DOC concentrations of 0.1 mg/L are quite low for detection on a Shimadzu TOC analyzer. Why were POC concentrations not measured directly rather than determined by loss on ignition? How large were PON concentrations? The filters were not acidified it appears so what about inorganic carbon?

Page 5, line 7: Where samples filtered through a smaller pore size filter than just a GF/F before optical analysis? In my experience, a 0.7 um filter does not remove enough of the background turbidity in low DOM, glacier water samples and therefore greatly interferes with the optical analysis. How many EEMs were included in the PARAFAC model? Given only 2 components were identified, I question the value of even including a PARAFAC model, especially given the uncertainties in sample processing and filtration.

Page 5, lines 21-22: Please provide some reporting/discussion of the physical data presented in Table 2 at some point?

Page 5, lines 24-29: If DOC concentrations in Iceland glaciers are comparable to other regions, do the authors have any idea of why POC concentrations are so different?

Page 5, lines 26-26: What were the % carbon concentrations on the filters and the TSS values? With such high POC concentrations, it would be helpful to see these data in a summary table or in Table 2.

Page 6, lines 11-12: It would be helpful to provide some more mechanistic information about how DOC and POC cycling is impacted by proglacial lakes. In other words, are biologic or physical processes impacting OC cycling?

Page 7, lines 1-2: If there is a strong anthropogenic influence at this site, not sure how much one can glean about OC dynamics and longitudinal changes in concentration

and speciation in proglacial streams? I think it is fine if the authors include this sample site in the results but remove this sample point when discussing longitudinal changes in proglacial streams.

Page 7, lines 3-5: I suggest the authors remove this sample point because of its salt-water influence. How can any conclusions be drawn about longitudinal changes in OC concentrations when a data point is influenced by saltwater rather than simply the fluvial network?

Page 7, line 23: Closer sampling points to what? The glacier terminus? Please clarify.

Page 7, lines 29-31: It is not clear how the authors make the link between fluorescence characteristics and ancient OC?

Page 8, line 5: This could be generally true but I am not convinced without some sort of regression or trendline with DOC/POC concentration versus distance from the glacier. There is a lot of longitudinal variability in OC concentration along the river. Moreover, some of these changes are likely driven by anthropogenic inputs and the influence of saltwater. So the trends (if any?) are not as simple and clear as stated here. I just plotted the DOC/POC data vs. distance from glacier terminus and found that DOC concentrations increased downstream. However, I found no trend what so ever for POC, especially once the last two data points (one with saltwater influence and the other with anthropogenic influence) were removed. The longitudinal approach needs to be revisited.

Page 8, lines 31-32: More detailed methods on the OC flux estimates are needed. What is the total runoff from the glaciers for the entire melt season? A mean runoff is not sufficient for estimating total annual OC fluxes from all of Iceland.

Page 9, lines 4-6: This is a very important methodological issue that should be addressed before this paper is considered for publication (see above). The POC concentrations presented here are not measured directly, highly variable and there are no

replicates (at least according the presented methods).

Page 9, lines 6-8: According to the Hood et al. (2015) paper, Icelandic glaciers are included in the estimates of global OC storage and release from glaciers and icefields?

Table 2: The pH and water temperature data do not seem realistic. I have never seen a pH value anywhere close to 13 in natural waters, even when water originates from limestone springs? Also, how can there be a stream temperature of 14C in a proglacial stream? A stream temperature of 5.6C 1km downstream from a glacier terminus? Are these sites receiving geothermal inputs of groundwater?

Figure 6: A regression plot with DOC/POC concentration vs. distance from glacier terminus would be more helpful. How were these "distance groupings" determined?

---

## Referee Comment (RC2) · Anonymous Referee #2 · 6 May 2018

The authors reported an organic carbon study in Iceland glaciers meltwaters, based on a field work in July of 2016. They reported important information for this island, especially for its POC and DOC concentration, as well as its DOC composition information. I think this is the most interesting key point for this study. However, I think the flaw is also obvious. I am writing the following suggestions for the authors to improve their manuscript. I think the major suggestions are the key problems that the current manuscript should overcome.

Major: 1. The POC measurement method. I think the authors did a very good job in DOC measurement. But for POC measurement, the method seems too old. As stated in the manuscript, there should be some interference (sometimes it may be very big) due to this old method. I think an elemental analyzer should be used, with

inorganic carbon being removed (e.g., via HCl) first. As POC data is widely presented and discussed in this manuscript, so I think this becomes a very clear flaw of the work. In addition, authors reported that POC flux in Iceland is very large in this work and take this as one of the key findings. Given the POC method problem, I think their suggestions (about the big POC flux) are not that convincing.

2. OC flux. In this ms, organic carbon flux is presented in the conclusion part, which is very strange. I think the flux estimate should be in a separate section, and with all the uncertainties presented and discussed.

3. Organic matter process in the glacier meltwaters. I got confused by the authors. At line 5 of page 7, authors suggest that the DOC concentration decrease is likely due to influence of seawater, as indicated by higher electrical conductivity. I went to Table 2, and check for the data of HV (ie. HV01-HV11). When I plot the conductivity against DOC concentration for all the HV station (ie. HV01-HV11), I found no such supporting relationship between conductivity and DOC concentration. Instead, DOC concentration seems slightly increase with increasing conductivity. This is in contrast with the authors words in line 5 of page 7. I have no idea if this is due to my mistake as I am not that familiar with the data as the authors are, but I think anyway the authors explanation here needs more attention.

4. POC decrease at line 7 page 8. I think authors should present stronger evidence to support their idea that the POC decrease may be due to direct use and reworking by benthic organisms, instead of citing a literature. Did they have evidence of benthic organisms in the sampling site? We indeed observed mosquito-like winged insect in some of the glacier meltwaters in the field. Sometimes there can be some other insects in some of the glacier meltwaters (benthic like). Is that the case in this current work? Are these insects being removed in the membranes before the sample was measured? On another aspect, authors would need evidence to prove that how significantly these author-mentioned microbial can contribute to modifying POC, given the short distance (and hence short time) and low temperature (and hence low rate) environment. I see

that DOC in glacier meltwater may be highly labile, but for POC, the condition can be quite different. I suggest the authors check if the POC variation is partly due to increasing conductivity or not. In addition, to present and discuss POC% data (POC in mg/L divided by TSM in mg/L) may be helpful too.

Minor: 1. about the length of the streams. In line 34 page 3 authors suggest that the streams are long. But later in line 20 page6, authors suggest that the streams is short. I got confused.

2. from line 10-20, page 3. The result from model predictions come with uncertainties. And so I suggest authors shorten the discussion and comparison for the glacier future changes.

3. the OC flux should be discussed with care. For example, OC may vary in both concentration and composition among different months within a melting season.

4. Figure 1. Please indicate the north direction.

---

## Author Comment (AC1) · 13 Jun 2018

We thank the anonymous referee #2 for careful reviewing our manuscript and for his constructive and interesting comments. We found these most helpful and have revised the manuscript accordingly. We are aware that there exist specific methodological uncertainties which have an impact especially on POC concentration and export estimations, but we would like to highlight, that this is an interesting pilot study for DOC and POC in Icelandic proglacial streams where none currently exists. We would like to point out that this study is an initial stepping stone that raises interesting questions from the observations and highlight the need to further investigations based on these initial studies.

[Figure]

Please find a formated version of this comment attached as .pdf.

Major comments:

(1) The POC measurement method. I think the authors did a very good job in DOC measurement. But for POC measurement, the method seems too old. As stated in the manuscript, there should be some interference (sometimes it may be very big) due to this old method. I think an elemental analyzer should be used, with inorganic carbon being removed (e.g., via HCl) first. As POC data is widely presented and discussed in this manuscript, so I think this becomes a very clear flaw of the work. In addition, authors reported that POC flux in Iceland is very large in this work and take this as one of the key findings. Given the POC method problem, I think their suggestions (about the big POC flux) are not that convincing.

Response:

The authors agree that the applied method for POC measurement (loss on ignition) is an older method and that it may be possible to eliminate several of the mentioned uncertainties by using an elemental analyzer. However, we are confident that we can account for the major sources of uncertainty in our current calculations. We are aware that clay minerals, present in volcanic ash and soils, may contain water of hydration, which is expelled at $\sim$300° (Lagaly 1993). The dehydration of typical volcanic ash hydrated clay minerals such as allophane may result in a weight loss of up to 36%, with most of the water of hydration lost at temperatures $\sim$ 110°C (Hensen and Smit 2002, Kitagawa 1972), similar to temperatures used for oven drying sediment. Other clay minerals such as kaolinite and montmorillonite show weight losses of about 14% and 15% respectively, losing most of their water of hydration at higher temperatures (Hensen and Smit 2002). Thus, accounting for the water of hydration within the respective sediment composition of our samples would allow for correction of current POC "over-estimates". Furthermore, other accepted methods for the determination of POC concentration can be used e.g., Federer et al. (2008) and Skidmore et al. (2000)

detected POC concentration by subtracting the DOC content of filtered water samples from the TOC content may be used to correct for uncertainties.

References:

Federer, U., Kaufmann, P. R., Hutterli, M. A., SchuÌĹpbach, S., & Stocker, T. F. (2008). Continuous flow analysis of total organic carbon in polar ice cores. Environmental Science & Technology, 42(21), 8039-8043. Hensen, E. J., & Smit, B. (2002). Why clays swell. The Journal of Physical Chemistry B, 106(49), 12664-12667. Kitagawa, Y. (1972): An aspect of the water in clay minerals: An application of nuclear magnetic resonance spectrometry to clay mineralogy. American Mineralogist 57:751-764 Skidmore, M. L., Foght, J. M. & Sharp, M. J. Microbial life beneath a high Arctic glacier. Appl. Environ. Microbiol. 66, 3214–3220 (2000).

(2) OC flux. In this manuscript, organic carbon flux is presented in the conclusion part, which is very strange. I think the flux estimate should be in a separate section, and with all the uncertainties presented and discussed.

Response: Thank you for pointing this out and for your suggestion. We have now added a separate paragraph to discuss DOC and POC fluxes and uncertainties.

(3) Organic matter process in the glacier meltwaters. I got confused by the authors. At line 5 of page 7, authors suggest that the DOC concentration decrease is likely due to influence of seawater, as indicated by higher electrical conductivity. I went to Table 2, and check for the data of HV (ie. HV01-HV11). When I plot the conductivity against DOC concentration for all the HV station (ie. HV01-HV11), I found no such supporting relationship between conductivity and DOC concentration. Instead, DOC concentration seems slightly increase with increasing conductivity. This is in contrast with the authors words in line 5 of page 7. I have no idea if this is due to my mistake as I am not that familiar with the data as the authors are, but I think anyway the authors explanation here needs more attention.

[Figure]

Response:

Thank you for highlighting the unclear wording. We have changed the wording to high-light the fact that we refer to the observed higher electrical conductivity as an indicator of streamwater-seawater mixing and the transition from stream to estuary, where other processes may influence DOC concentration at this sampling point. "At the sampling point HV01, located at the end of the lagoon Ölfusá, in the estuary of the river Hvitá, the DOC concentration decreased, possibly as a result of the influence of incoming seawater, indicated by the substantially higher electrical conductivity at this location (Table 2)."

(4) POC decrease at line 7 page 8. I think authors should present stronger evidence to support their idea that the POC decrease may be due to direct use and reworking by benthic organisms, instead of citing a literature. Did they have evidence of benthic organisms in the sampling site? We indeed observed mosquito-like winged insect in some of the glacier meltwaters in the field. Sometimes there can be some other insects in some of the glacier meltwaters (benthic like). Is that the case in this current work? Are these insects being removed in the membranes before the sample was measured?

Response:

The authors agree that insects which are released on the filters can influence the mea-surements of POC concentration. To avoid such impacts, the authors checked every GF/F filter directly after the sampling and proceeding measurements with no indication of insects on the filters. Additionally, water samples were taken in the upper part of the water column reducing the likelihood of capturing benthic organisms during sampling. Furthermore, POC retention is strongly correlated with macroinvertebrate abundances, particularly with respect to detritivorous invertebrates which utilize POC as source of nutrition (Dangles et al. 2001; Monaghan et al. 2001). POC will be directly used by a variety of invertebrates and so may represent a more direct pathway of carbon trans-fer than the "microbial loop" with its inherent respiratory losses (Findlay 1995). The

findings of Gislason et al. (2001) and Lods‐Crozet et al. (2001) support the idea that the observed POC decrease may be due to direct use and reworking by benthic organisms. Here, filter-feeders (especially Simulidae, blackflies) of POC only occur in downstream reaches of the glacial river, while the most upper river sections are only inhabited by algae scrapping chironomid larvae due to the harsh environmental conditions.

Dangles, O., Guerold, F., Usseglio‐Polatera, P. (2001). Role of transported particulate organic matter in the macroinvertebrate colonization of litter bags in streams. Freshwater Biology, 46(5), 575-586. Monaghan, M. T., Thomas, S. A., Minshall, G. W., Newbold, J. D., Cushing, C. E. (2001). The influence of filter‐feeding benthic macroinvertebrates on the transport and deposition of particulate organic matter and diatoms in two streams. Limnology and Oceanography, 46(5), 1091-1099. Findlay, S. (1995). Importance of surface‐subsurface exchange in stream ecosystems: The hyporheic zone. Limnology and oceanography, 40(1), 159-164. Gíslason, G. M., Adalsteinsson, H., Hansen, I., Ólafsson, J. S., Svavarsdóttir, K. (2001). Longitudinal changes in macroinvertebrate assemblages along a glacial river system in central Iceland. Freshwater Biology, 46(12), 1737-1751. Lods‐Crozet, B., Lencioni, V., Olafsson, J. S., Snook, D. L., Velle, G., Brittain, J. E. & Rossaro, B. (2001). Chironomid (Diptera: Chironomidae) communities in six European glacier‐fed streams. Freshwater Biology, 46(12), 1791-1809.

(5) On another aspect, authors would need evidence to prove that how significantly these author-mentioned microbial can contribute to modifying POC, given the short distance (and hence short time) and low temperature (and hence low rate) environment. I see that DOC in glacier meltwater may be highly labile, but for POC, the condition can be quite different. I suggest the authors check if the POC variation is partly due to increasing conductivity or not. In addition, to present and discuss POC% data (POC in mg/L divided by TSM in mg/L) may be helpful too.

Response:

Thank you for your suggestion. We agree, that microbes may not rework such high amounts of POC within such a short distance, but rather DOC. Presently, in the Icelandic glacier-fed streams, microbial reworking of POC is but a first attempt to explain the decrease of POC concentration. This aquatic process has been already identified in similar environments. In fact, POC has been shown to be directly used by a variety of invertebrates and thus may represent a more direct pathway of carbon transfer than the "microbial loop" with its inherent respiratory losses (Findlay 1995). As mentioned above, the results by Gíslason et al. (2001) and Lods‐Crozet et al. (2001) supporting the evidence of the idea that the POC decrease may be due to direct use and reworking by benthic organisms. Here, filter-feeders (especially Simulidae, black flys) of POC only occur in downstream reaches of the glacial river and the most upper parts are only inhabited by algae scrapping chironomid larvae due to the harsh environmental conditions. Thus, the authors suggest this to be a first possible explanation with the recommendation for further future research in these glacial-fed streams. We will re-word the text to clarify this point.

"As a first hypothesis, we suggest that the observed decrease in POC may be due to direct use and reworking by benthic organisms (Findlay, 1995) and recommend further investigation into the rapid decrease in POC over relatively short river lengths in Icelandic streams. At the same time, DOM composition changed from relatively more allochthonous to autochthonous DOM."

Minor comments:

(1) About the length of the streams. In line 34 page 3 authors suggest that the streams are long. But later in line 20 page 6, authors suggest that the streams is short. I got confused.

Response:

Thank you for you for highlighting this. In line 20 of page 6, the length of the proglacial streams refer exclusively to the proglacial streams draining the southern parts of the

glaciers Mýrdalsjökull and Vatnajökull. At page 3, line 34, the authors mention that in comparison to Greenland and additionally to the proglacial streams in the southern part of the two glaciers, there exist several longer proglacial streams which are draining e.g., the northern parts of Hofsjökull and Vatnajökull. Thus, both possibilities are given in Iceland. The manuscript text will be adjusted to further clarify these. "Such elevated concentrations may have implications for biogeochemical processes in the ocean, due to the relatively short distance from the glacier terminus of the Mýrdalsjökull and Vatnajökull glaciers to the North Atlantic Ocean compared to the other glaciers (Bhatia et al., 2013)"

(2) From line 10-20, page 3. The result from model predictions come with uncertainties. And so I suggest authors shorten the discussion and comparison for the glacier future changes.

Response:

Thank you for your comment. We agree that model predictions always come with specific uncertainties. In this section we aimed to highlight the common future development of the shrinking of Icelandic glaciers and not in reference to the estimation of the DOC and POC export. We believe that the present text is required to stress this point of importance of the study given current predictions for the various Icelandic glaciers.

(3) The OC flux should be discussed with care. For example, OC may vary in both concentration and composition among different months within a melting season.

Response:

Thank you for your comment. We agree that there may be considerable variation in OC concentration and composition within the melting period and thus our estimates taken during peak melt season may lead to high OC fluxes. Therefore, we now discuss the seasonal variation in OC and that the amount of both dissolved and particulate organic carbon may be in fact, on average, lower than we estimate from our study during peak

melt.

(4) Figure 1. Please indicate the north direction.

Response:

Thank you for highlighting this omission. We have now inserted a north arrow as suggested.

Please also note the supplement to this comment:
https://www.the-cryosphere-discuss.net/tc-2018-32/tc-2018-32-AC1-supplement.pdf

---

## Author Comment (AC2) · 13 Jun 2018

We thank the anonymous referee #1 for careful reviewing our manuscript and for his constructive and interesting comments. We found these most helpful and have revised the manuscript accordingly. We are aware that there exist specific methodological uncertainties which have an impact especially on POC concentration and export estimations, but we would like to highlight, that this is an interesting pilot study for DOC and POC in Icelandic proglacial streams where none currently exists. We would like to point out that this study is an initial stepping stone that raises interesting questions from the observations and highlight the need to further investigations based on these initial studies.

General comments:

(1) There are far too many methodological uncertainties/limitations for consideration of publication. For instance, a great deal more information is needed for sample processing and analysis, OC flux estimates, and assessment of analytical error.

Response:

We thank the reviewer for pointing this out. We have included more information on sample processing and analysis in the manuscript. For the analyses of POC and DOC concentration and optical analysis, the water samples (150 ml) were filtered through a double layer of pre-combusted (450° C) glass fibre filters (Whatman GF/F, pore size 0.7 $\mu$m) according to Singer et al. (2012). The filters were stored in an aluminum paper separately and kept cool and dark. Samples for DOC and optical analysis were stored in 40 ml glass vials (soaked with 0.1 N HCl, rinsed thoroughly with purified water and combusted for 4h at 450°C), sealed with Teflon-coated septa (soaked with 0.1 N NaOH and rinsed thoroughly with purified water). The water samples were stored in a dark cooling box until shipment and laboratory analysis.

POC was measured by determining mass lost upon combustion of the samples. The glass fiber filters were dried after sampling at 65°C to a constant weight to determine the total suspended solids (TSS). The samples were then combusted at 550° C and re-weighed to calculate the amount of particulate organic matter according to Maciejewska and Pempkowiak (2014). We are aware that clay minerals, present in volcanic ash and soils, may contain water stored within the layers. This water of hydration may influence the amount of particulate organic matter since it will be expelled at $\sim$300° (Lagaly 1993) during combustion of the sample (550° C). The dehydration of hydrated clay minerals such as allophane, which are typical of volcanic ash, may result in a weight loss of up to 36%, with the majority of water of hydration being lost at $\sim$ 110°C (Hensen and Smit 2002, Kitagawa 1972). Other common clay minerals such as kaolinite or montmorillonite show weight losses of about 14% and 15%, but they lose the most part of

water only at higher temperatures (Hensen and Smit 2002). Thus, accounting for the water of hydration within the respective sediment composition of our samples would allow for correction of current POC "over-estimates".

DOC concentrations were measured using a TOC analyzer (TOC-L, Shimadzu, Japan) using high-temperature combustion of organic matter (OM) followed by thermal detection of $CO_2$. Using this method, the detection limit of DOC is at 4 $\mu$g/L.

We estimated fluxes using a very simple approach based on annual glacial discharges (Björnsson and Pálsson 2008) and measured POC/DOC concentrations in the proglacial streams nearby the glacier termini. Although it is likely that a substantial amount of the measured POC/DOC concentrations originates from the glaciers, we acknowledge that we cannot directly infer loss from glaciers. We have carefully rephrased the manuscript text to reflect this fact.

Björnsson, H. and Pálsson, F. (2008): Icelandic glaciers, Jökull, 58:365–386. Hensen, E. J., & Smit, B. (2002). Why clays swell. The Journal of Physical Chemistry B, 106(49), 12664-12667. Kitagawa, Y. (1972): An aspect of the water in clay minerals: An application of nuclear magnetic resonance spectrometry to clay mineralogy. American Mineralogist 57:751-764

(2) POC concentrations are not measured directly but rather by loss on ignition.

a. First off, this is not a very commonly used technique for POC analysis, especially in stream biogeochemistry.

b. Second, the authors clearly acknowledge that hydrated clay minerals are likely contributing to the high POC concentrations.

c. Third, it is clear that the POC concentrations are highly variable and greatly overestimated and how can concentrations increase from <1 to 56 mg/L over a distance of 3 km in river length?

d. Without any replicates or lab studies comparing direct POC vs. LOI analyses, I don't

know how these data can be compared to other glacier OC studies with direct POC analysis.

Response:

a) Thank you for your comments. The authors agree that the applied method for POC measurement (loss on ignition) is an older method and that it may be possible to eliminate several of the mentioned uncertainties by using an elemental analyzer. However, we are confident that we can account for the major sources of uncertainty in our current calculations such as loss of weight by water stored in clay minerals.

b) As mentioned above the dehydration of hydrated clay minerals like allophane, which are typical minerals in volcanic ash, may result in a weight loss of up to 36%, with most loss occuring at $\sim 110°C$ (Hensen and Smit 2002, Kitagawa 1972). Other clay minerals like kaolinite or montmorillonite show weight losses of about 14% and 15%. Thus, accounting for the water of hydration within the respective sediment composition of our samples would allow for correction of current POC "over-estimates". Additionally, recently Eiriksdottir et al. (2017) calculated annual POC flux for the two rivers Jökulsá á Dal and Lagafljót in the north of Vatnajökull which can be used for comparison. Furthermore, other accepted methods for the determination of POC concentration can be used e.g., Federer et al. (2008) and Skidmore et al. (2000) detected POC concentration by subtracting the DOC content of filtered water samples from the TOC content may be used to correct for uncertainties.

Eiriksdottir, E. S., Oelkers, E. H., Hardardottir, J., & Gislason, S. R. (2017): The impact of damming on riverine fluxes to the ocean: A case study from Eastern Iceland. Water research, 113, 124-138.

c) The authors agree that the measured POC concentrations are very variable in the Icelandic proglacial streams. This may due to different factors which are specific for each stream and the corresponding glacier. We acknowledge that the specific sources of POC have to be investigated in detail, which could not be achieved in the present

study, but will be part of a future research project (recently accepted by the German Research Foundation), we have highlighted this fact in the revised manuscript.

d) Taking all the methodological uncertainties analyzing POC in this manuscript into account the authors intend for focus to be placed not the annual estimations of POC export but rather the spatial variability of the POC concentrations. We have changed the manuscript to reflect this aspect.

Specific comments:

(1) Page 2, lines 20-22: This is confusing. Icelandic glaciers are included in the global estimates of DOC and POC release from glaciers. However, as the authors point out, concentration estimates of OC in Icelandic glaciers as well as many other regions worldwide are not. Please rephrase to improve clarity.

Response:

Thank you for highlighting this. We have corrected the sentence to avoid confusion: Estimations of the global release of DOC and POC by Hood et al. (2015) are based solely on 23 samples of the Antarctic Ice Sheet, 9 samples of the Greenland Ice Sheet and 55 samples of mountain glaciers. Furthermore, although the release of glacial organic carbon has been investigated in proglacial streams and in the glacial ice in Alaska (Spencer et al., 2014; Hood et al., 2015), the European Alps (Singer at al., 2012), Greenland (e.g., Bhatia et al., 2010; 2011; 2013; Lawson et al., 2014), Svalbard (e.g., Zhu et al., 2016), and Asia (e.g., Spencer et al., 2014), to our knowledge, there is no comparable information available for Iceland. Thus, Icelandic glaciers are not included in the derivation of the global release of DOC and POC by Hood et al. (2015) which is surprising as the largest nonpolar ice cap of Europe (Vatnajökull) is located in Iceland (Björnson et al., 2013).

(2) Page 4, line 10: I think the authors have already made it clear that there are no other studies, to their knowledge, of OC release from Iceland glaciers.

Response:

Thank you for highlighting this repetition. We have deleted this sentence.

(3) Page 4, line 14: This is a very short sampling window. How long does the melt season last? When is the peak of the melt season? How many times was each stream sampled?

Response:

Thank you for highlighting this unclear statement. A key objective of this study was to obtain an insight into the DOC and POC fluxes especially during the peak melt season where we expect the highest concentrations in comparison to the non-melt season. Thus, if concentration would be low, in comparison with other studies described in Hood et al. (2015), the contribution of glacial organic carbon of Icelandic glaciers to the global release would be low, too, and further studies not necessary. But the results of this first initial study show an important release of DOC and POC which makes further studies necessary to enhance the global estimations. The melting season typically lasts from May to November with the peak melting season occurring over 6 weeks, typically in the month of July and August. As this was a pilot study as described above, we sampled only 1 times per stream. We have now included these details in the manuscript to ensure clarity.

"We sampled stream water from 25 sites, draining a total of 5 Icelandic glaciers over a week (23-31 July 2016) during the peak of the melting season. The melting season was chosen for sampling as during this period the ablation zones of the glaciers are free of snow and the proglacial streams cover OC of different meltwater sources (supraglacial, englacial and subglacial) according to the findings of Bhatia et al. (2011; 2013) and Das et al. (2008)"

(4) Page 4, lines 26-29: Given the extremely low DOC concentration reported here, more information is needed on sample handling and processing. For instance, were

samples field filtered or acidified in the field? Were replicate samples collected? How long were sampled stored in the field before transport? How long until analysis for DOC occurred?

Response:

Thank you for highlighting these missing details. We have now included the details below: For the analyses of DOC concentration, the water samples (150 ml) were filtered through a double layer of pre-combusted (450° C) glass fibre filters (Whatman GF/F, pore size 0.7 $\mu$m) according to Singer et al. (2012) without replicates. Samples for DOC were stored in 40 ml glass vials (soaked with 0.1 N HCl, rinsed thoroughly with purified water and combusted for 4h at 450°C), sealed with Teflon-coated septa (soaked with 0.1 N NaOH and rinsed thoroughly with purified water). We did not acidify the samples as this may affect EEM properties. However we fixed the samples using a double layer of GF/F filters and stored them at 4°C in the dark until further analysis (Hudson et al. 2007, Donahue et al. 1998). The water samples were stored in a dark cooling box until shipment and laboratory analysis. The lab analyses started immediately after the return from Iceland to Germany, thus, first water samples were analyzed 8 days after sampling. Hudson, N., Baker, A., & Reynolds, D. (2007). Fluorescence analysis of dissolved organic matter in natural, waste and polluted waters—a review. River Research and Applications, 23(6), 631-649. Donahue, W. F., Schindler, D. W., Page, S. J., & Stainton, M. P. (1998). Acid-induced changes in DOC quality in an experimental whole-lake manipulation. Environmental science & technology, 32(19), 2954-2960.

(5) Page 5, lines 1-6: More information is needed about the DOC and POC analysis...

a. Were replicated analyses performed?

b. What is the lower detection for DOC? Are there any error estimates on the OC concentrations? DOC concentrations of 0.1 mg/L are quite low for detection on a Shimadzu TOC analyzer.

c. Why were POC concentrations not measured directly rather than determined by loss on ignition?

d. How large were PON concentrations?

e. The filters were not acidified it appears so what about inorganic carbon?

Response:

Thank you for your questions and for highlighting the need for clarification on these issues.

a) No replicates were performed, but we present a relatively high number of consecutive downstream samples

b) DOC concentrations were measured using a TOC analyzer (TOC-L, Shimadzu, Japan) using high-temperature combustion of organic matter (OM) followed by thermal detection of $CO_2$. Using this method the detection limit of DOC is indicated at 4 $\mu$g/L. Prior to injection, DOC samples (GF/F-filtered) were automatically acidified in the analyzer as recommended by the manufacturer. Using purified (Milli-Q) water, we also determined blanks for the determination of DOC concentration.

c) For this initial field study we decided to measure POC concentrations by loss of ignition since in the light of all uncertainties this is a common and well known method. Although POC via loss of ignition is an often applied method, we acknowledge the high uncertainties related to this method. However our POC measurements are closely bracketed by measurements of POC by Kardjilov et al. 2006 and Eiriksdottir et al. 2017.

Kardjilov, M. I., Gisladottir, G., & Gislason, S. R. (2006). Land degradation in north-eastern Iceland: present and past carbon fluxes. Land degradation & development, 17(4), 401-417.

d) As our study was an initial study we did not measure PON concentrations, but we

would like to refer to Eiriksdottir et al. (2017) who detected a mean annual flux of 185 ton/yr and 106 ton/year (period 1998-2003), respectively, in the two rivers Jökulsá á Dal and Lagafljót in the north of Vatnajökull. We recognize the importance of this and have highlighted this in the manuscript.

e) As we measured POC we did not acidify the filters as inorganic carbon is combusted at higher temperatures than organic carbon.

(6) Page 5, line 7: Where samples filtered through a smaller pore size filter than just a GF/F before optical analysis? In my experience, a 0.7 um filter does not remove enough of the background turbidity in low DOM, glacier water samples and therefore greatly interferes with the optical analysis. How many EEMs were included in the PARAFAC model? Given only 2 components were identified, I question the value of even including a PARAFAC model, especially given the uncertainties in sample processing and filtration.

Response:

Thank you for your comments. The water samples were filtered through a double layer of GF/F filters directly after the sampling, but not through a smaller pore size filter. Using GF/F filters and the pore sizes of 0.7 $\mu$m is a common method within organic carbon research to separate particulate and dissolved contents and to detect optical properties, applied by e.g., Foreman (2017), Paulsen et al. (2017), Singer et al. (2012), Skidmore et al. (2000) or Smith et al. (2017).

Foreman, C. M. (2017). Microbial formation of labile organic carbon in Antarctic glacial environments. Paulsen, M. L., Nielsen, S. E., Müller, O., Møller, E. F., Stedmon, C. A., Juul-Pedersen, T., Markager, S., Sejr, M.K., Huertas, A.D., Larsen, A., Middelboe, M. (2017). Carbon bioavailability in a high Arctic fjord influenced by glacial meltwater, NE Greenland. Frontiers in Marine Science, 4, 176. Singer, G. A., Fasching, C., Wilhelm, L., Niggemann, J., Steier, P., Dittmar, T., & Battin, T. J. (2012). Biogeochemically diverse organic matter in Alpine glaciers and its downstream fate. Nature Geoscience,

5(10), 710. Skidmore, M. L., Foght, J. M. & Sharp, M. J. Microbial life beneath a high Arctic glacier. Appl. Environ. Microbiol. 66, 3214–3220 (2000). Smith, H. J., Foster, R. A., McKnight, D. M., Lisle, J. T., Littmann, S., Kuypers, M. M., & Foreman, C. M. (2017). Microbial formation of labile organic carbon in Antarctic glacial environments. Nature Geoscience, 10(5), 356-359.

(7) Page 5, lines 21-22: Please provide some reporting/discussion of the physical data presented in Table 2 at some point?

Response:

Thank you for your suggestion. We have included a brief discussion of the physical parameter spatial variability with reference to Table 2, in the results and discussion sections.

(8) Page 5, lines 24-29: If DOC concentrations in Iceland glaciers are comparable to other regions, do the authors have any idea of why POC concentrations are so different?

Response:

Thank you for your comment. We acknowledge the substantial difference between DOC and POC concentrations. We suggest that the local conditions receiving allochthonous POC stored in the glacier could be one reason for this, an aspect that needs to be further investigated. Ideally, further studies with respect to the determination of the molecular composition of the glacial derived organic carbon could give a more detailed insight into the source of POC and further, the reason why DOC and POC concentration are so different. We have highlighted these now in the manuscript text.

(9) Page 5, lines 26-26: What were the % carbon concentrations on the filters and the TSS values? With such high POC concentrations, it would be helpful to see these data in a summary table or in Table 2.

Response:

Thank you for this suggestion. We will add the TSS values to the Table 2.

(10) Page 6, lines 11-12: It would be helpful to provide some more mechanistic information about how DOC and POC cycling is impacted by proglacial lakes. In other words, are biologic or physical processes impacting OC cycling?

Response:

Thank you for suggesting this. We have now further elaborated on this in the manuscript. Indeed, different mechanisms impact DOC and POC in proglacial lakes (Sommaruga et al 2001). Especially turbidity, a direct consequence of glacial milk, impacts UV attenuation as well as microbial community composition and function (Peter and Sommaruga 2016). Furthermore sedimentation may play a role. Peter, H., & Sommaruga, R. (2016). Shifts in diversity and function of lake bacterial communities upon glacier retreat. The ISME journal, 10(7), 1545.

(11) Page 7, lines 1-2: If there is a strong anthropogenic influence at this site, not sure how much one can glean about OC dynamics and longitudinal changes in concentration and speciation in proglacial streams? I think it is fine if the authors include this sample site in the results but remove this sample point when discussing longitudinal changes in proglacial streams.

Response:

Thank you for your comment. In the present manuscript we did not include this sampling point into the calculation of the DOC and POC release of Icelandic glaciers, since at this sampling point the human impact is obvious due to the adjacent village. But, we believe that this sampling point is valid in terms of the delivery of OC to the ocean and variability of OC quality and in the discussion of longitudinal changes (both natural and anthropogenic) in proglacial streams.

(12) Page 7, lines 3-5: I suggest the authors remove this sample point because of its

saltwater influence. How can any conclusions be drawn about longitudinal changes in OC concentrations when a data point is influenced by saltwater rather than simply the fluvial network?

Response:

Thank you for your comment. As mentioned above we also want to understand how much and what OC in terms of quality reaches the Atlantic Ocean, monitoring changes along the entire continuum. Therefore we believe that the knowledge about DOC and POC concentration after the lagoon is important to mention (acknowledging the fact that it is influenced by seawater).

(13) Page 7, line 23: Closer sampling points to what? The glacier terminus? Please clarify.

Response:

Thank you for highlighting this. We have clarified this sentence: "Although glacial DOM generally exhibited high proteinaceous fluorescence (C2), the PCA revealed DOM properties to vary among glaciers, with sampling points closer to the glacier termini being more closely related in terms of DOM optical composition. "

(14) Page 7, lines 29-31: It is not clear how the authors make the link between fluorescence characteristics and ancient OC?

Response:

Thank you for pointing this out. We have rephrased this section carefully taking into account that the humic-like fluorescence may originate from multiple sources, like surrounding soils, ancient vegetation. Supporting this notion Singer et al (2012) found phenolic substances in glacial ice likely originating from ancient vegetation.

(15) Page 8, line 5: This could be generally true but I am not convinced without some sort of regression or trendline with DOC/POC concentration versus distance from the

glacier. There is a lot of longitudinal variability in OC concentration along the river. Moreover, some of these changes are likely driven by anthropogenic inputs and the influence of saltwater. So the trends (if any?) are not as simple and clear as stated here. I just plotted the DOC/POC data vs. distance from glacier terminus and found that DOC concentrations increased downstream. However, I found no trend what so ever for POC, especially once the last two data points (one with saltwater influence and the other with anthropogenic influence) were removed. The longitudinal approach needs to be revisited.

Response:

Thank you for your comments and raising the need for clarification. We agree that there exist a high variability in POC concentration along the river Hvitá and the concluded mean "trend" is not clear due to the variability and therefore, cannot be captured by a formula, but this was not the intended objective. Reasons for the variability are manyfold for varying streams, and we discuss possible reasons here in this first study, highlighting the need for further investigations where needed. Concerning the increase of DOC concentration along the river Hvrità, the authors described it accurately in line 5 of page 8 that. We found POC to generally decrease, while DOC increased with the distance from the glacier termini 5 (Figure 6). In this context reference should be changed to Figure 2 and we will clarify that the trend refers solely to the river Hvitá, the site with the most continuous downstream sampling points.

(16) Page 8, lines 31-32: More detailed methods on the OC flux estimates are needed. What is the total runoff from the glaciers for the entire melt season? A mean runoff is not sufficient for estimating total annual OC fluxes from all of Iceland.

Response:

Thank you for this comment. We are aware of this fact and highlight that this study is very first rough estimation of the annual OC fluxes with different uncertainties and we are fully aware that for a detailed estimation more water samples during different runoff

conditions are necessary. In this initial study a key goal was to obtain a first insight into the DOC and POC fluxes, where none exists, especially during the peak melt season where we expect the highest concentrations in comparison to the non-melt season. Similar to other studies e.g., Singer et al. (2012) where single investigations are the basis for OC flux estimation we applied this approach to Iceland. Of course Singer et al. (2012) used the annual mass balance for flux estimation, but ice samples were also taken only during the peak melt season. We will carefully rephrase the manuscript text to reflect these points and the need for further investigation to constrain possible errors and obtain improved estimates.

(17) Page 9, lines 4-6: This is a very important methodological issue that should be addressed before this paper is considered for publication (see above). The POC concentrations presented here are not measured directly, highly variable and there are no replicates (at least according the presented methods).

Response:

Thank you for your comments. As mentioned above, this initial study where the key goal was to obtain a first insight into the DOC and POC fluxes, where none currently exists, especially during the peak melt season where we expect the highest concentrations in comparison to the non-melt season. We agree that detection of POC concentration by combustion is an old method with associated inaccuracies. However, we have taken measures to account for specific uncertainties with respect to loss of water of hydration from sediments and included more information about the flux estimation into the manuscript.

(18) Page 9, lines 6-8: According to the Hood et al. (2015) paper, Icelandic glaciers are included in the estimates of global OC storage and release from glaciers and icefields?

Response:

We apologize for this confusing sentence, which we have now corrected from: "Nev-

ertheless, compared to the global release of 1.97 Tg C yr-1 (POC) estimated by Hood et al. (2015), these first calculations underline the absolute necessity to include the Icelandic glaciers in the derivation of global organic carbon budgets."

to:

"The estimations of the global release of DOC and POC by Hood et al. (2015) are based only on 23 samples of the Antarctic Ice Sheet, 9 samples of the Greenland Ice Sheet and 55 of mountain glaciers but Icelandic glaciers are not included in the derivation of this estimation."

(19) Table 2: The pH and water temperature data do not seem realistic. I have never seen a pH value anywhere close to 13 in natural waters, even when water originates from limestone springs? Also, how can there be a stream temperature of 14C in a proglacial stream? A stream temperature of 5.6C 1km downstream from a glacier terminus? Are these sites receiving geothermal inputs of groundwater?

Response:

We agree that the pH-values as stated are very high and it is likely that there was an error in transcription or error in calibration of the device. We will investigate and address accordingly the incidence of high pH in the revised manuscript. Concerning the water temperature input of geothermal water could be possible.

(20) Figure 6 (Please find this figure in the attached .pdf-file): A regression plot with DOC/POC concentration vs. distance from glacier terminus would be more helpful. How were these "distance groupings" determined?

Response:

Thank you for this suggestion. We added a regression of the POC/DOC ratio and the distance from the glacier terminus (r2=0.44, p<0.001, n=23). The distance groupings were made in such a manner to consider sites of similar distances from the various investigated rivers to present different typical distances and in consideration of the

different types of rivers: glaciated vs unglaciated

Please also note the supplement to this comment:
https://www.the-cryosphere-discuss.net/tc-2018-32/tc-2018-32-AC2-supplement.pdf
* * *